# Influence of Triazole Pesticides on Wine Flavor and Quality Based on Multidimensional Analysis Technology

**DOI:** 10.3390/molecules25235596

**Published:** 2020-11-28

**Authors:** Ouli Xiao, Minmin Li, Jieyin Chen, Ruixing Li, Rui Quan, Zezhou Zhang, Zhiqiang Kong, Xiaofeng Dai

**Affiliations:** 1Feed Research Institute, Chinese Academy of Agricultural Sciences, Beijing 100081, China; xiaoouli123@163.com (O.X.); zhangzezhou7689@163.com (Z.Z.); 2State Key Laboratory for Biology of Plant Diseases and Insect Pests, Institute of Plant Protection, Chinese Academy of Agricultural Sciences, Beijing 100193, China; chenjieyin@caas.cn; 3Key Laboratory of Agro-Products Quality and Safety Control in Storage and Transport Process, Ministry of Agriculture and Rural Affairs/Institute of Food Science and Technology, Chinese Academy of Agricultural Sciences, Beijing 100193, China; liminmin@caas.cn (M.L.); liruixing06@163.com (R.L.); qr802319@163.com (R.Q.)

**Keywords:** triazole pesticides, wine, fermentation, sensory analysis, flavor components

## Abstract

Triazole pesticides are widely used to control grapevine diseases. In this study, we investigated the impact of three triazole pesticides—triadimefon, tebuconazole, and paclobutrazol—on the concentrations of wine aroma compounds. All three triazole pesticides significantly affected the ester and acid aroma components. Among them, paclobutrazol exhibited the greatest negative influence on the wine aroma quality through its effect on the ester and acid aroma substances, followed by tebuconazole and triadimefon. Qualitative and quantitative analysis by solid-phase micro-extraction gas chromatography coupled with mass spectrometry revealed that the triazole pesticides also changed the flower and fruit flavor component contents of the wines. This was attributed to changes in the yeast fermentation activity caused by the pesticide residues. The study reveals that triazole pesticides negatively impact on the volatile composition of wines with a potential undesirable effect on wine quality, underlining the desirability of stricter control by the food industry over pesticide residues in winemaking.

## 1. Introduction

Wine has become one of the three most globally popular alcoholic beverages because of its good flavor and taste and its unique health benefits. Indeed, a decreased risk of cardiovascular disease, improved immunity, and reduced mortality rate have been reported in moderate wine drinkers [1]. According to statistics, with grapes being the raw material in wine production, more than half of the global annual grape production is used for wine production [2]. However, because of their high fructose and glucose contents, grapes are susceptible to contamination by vine microbial pathogens during cultivation. Thus, chemical pesticides are applied to control diseases and pests during the whole grape cultivation cycle to obtain high-quality wine grapes [3]. Unfortunately, unsuitable agricultural practices associated with potential grape and wine contamination, are frequently observed during the application of these active materials [4]. Consequently, consumers are indirectly exposed to these pesticides, which pose potential risks to human health. Thus, there is an ever-increasing global concern about wine quality and food safety.

Triadimefon [1-(4-chlorophenoxy)-3,3-dimethyl-1-(1*H*-1,2,4triazol-1-yl)butan-2-one], tebuconazole [(*RS*)-1-(4-chlorophenyl)-4,4-dimethyl-3-(1*H*-1,2,4-triazol-1-ylmethyl)-pentan-3-ol], and paclobutrazol [(2*RS*,3*RS*)-1-(4-chlorophenyl)4,4-dimethyl-2-(1*H*-1,2,4-triazol-1-yl)pentan-3-ol] (Appendix A) are triazole pesticides commonly used in grape cultivation [5]. These pesticides (e.g., triadimefon and tebuconazole) are mainly used as fungicides because of their broad-spectrum activity and protective, curative, and eradicating action against actinomycetes and basidiomycetes [6], such as *Streptomyces scabies*, a genus of actinomycetes, which causes scabs in tap root crops and potato tubers [7]. Moreover, the grape skin extract containing fungicide can obviously inhibit the germination of *Penicillium expansum*, *Penicillium chrysogenum*, and *Aspergillus niger* [8]. However, some triazole pesticides (e.g., paclobutrazol) are also used to regulate plant extension growth. The Joint FAO (Food and Agriculture Organization of the United Nations)/WHO (World Health Organization) Meeting on Pesticide Residues (JMPR), reported the data from 17 residue trials matching good agricultural practice (GAP), with the highest pesticide residue of 0.52 mg/kg in grapes. It was also reported that the use of triadimefon on grapes would contribute to high pesticide residues of 3.2 mg/kg [9]. Additionally, many triazole pesticides have been reported as potential endocrine disruptors with anti-androgen activity and inhibition of the enzymatic activity of cytochrome 3A4 (CYP3A4), which was the dominant form of CYP450 in the liver that mediates the 6b-hydroxylation of testosterone [10]. Thus, it is important to monitor the presence of pesticides and regulate their levels in grapes and wine to limit human health risks, and to use phytochemical biopesticides that are less toxic, least persistent, environmentally friendly, and safe to humans and non-target organisms. It was reported that several phytochemical biopesticides such as azadirachtin, nicotine, pyrethrins, rotenone, veratrum, annonins, rocaglamides, isobutylamides, etc. have been successfully commercialized in the past [11].

Maximum residue levels (MRLs) in plant-based products have been established by many countries and international organizations to regulate pesticide levels. The MRLs for pesticide residues in grapes are normally in the range of 0.01–5 mg/kg, depending on the pesticides [12]. For example, in China, the MRLs of triadimefon, tebuconazole, and paclobutrazol on grapes are 0.3, 2.0, and 0.5 mg/kg, respectively [13]. In addition, pesticides that penetrate plant tissue and contaminate the grape will lead to stimulated or sluggish wine fermentation [14]. Moreover, the residual pesticides on grapes affect the flavor characteristics of wine volatile compounds through fermentation [15]. In summary, these pesticide residues not only pose potential health risks to the consumers but also reduce the wine quality.

Red wine is fermented with the existing pericarp to extract more chemical components (e.g., anthocyanins, polyphenols, and volatile compounds). This is because the wine aroma originates from the substances produced by the fermentation of the grape itself and yeast. González-Álvarez et al. [16] reported that residual fungicides affected the contents of discriminant volatiles (4-vinylguaiacol, 3-methylbutanoic acid and acetates, and ethyl ester) produced by biosynthesis using fatty acids as precursors, thereby changing the flavor of white wine. Pesticides can also affect the activity of lactic acid bacteria and change the process of malolactic fermentation [17]. Notably, flavor substance is an important quality index to judge the quality of wine as well as an important factor that affects the purchase intention of consumers [18].

Instrumental analytical technologies have attracted much attention in the analysis and identification of volatile components in food because of their objectivity, high efficiency, sensitivity, and stability. Particularly, electronic nose and tongue technologies are intelligent smell and taste recognition systems that simulate the respective human physiological senses [19]. These techniques are commonly applied to the flavor analysis of various food products such as beverages (e.g., wine) and condiments [20]. The combination of electronic nose and tongue sensing makes the detection result more accurate. Solid-phase micro-extraction gas chromatography coupled with mass spectrometry (SPME-GC-MS) and gas chromatography-ion mobility spectrometry (GC-IMS) have been widely used in food flavor analysis. Applications include the analysis of volatile components in different vintages [21], comparison of different fermentation properties of raspberry wine [22], etc. These instrumental analytical technologies can effectively separate and identify the volatile compounds responsible for the fragrance of the product. Moreover, the application of these techniques can effectively avoid the risk of exposing traditional human sensory evaluators to the chemical hazards present in contaminated target samples. To the best of our knowledge, literature on the influence of triazole pesticides on wine flavor and quality based on electronic sensory evaluation systems and GC-MS is scarce [15].

With these facts in mind, we aimed to investigate the effects of triadimefon, tebuconazole, and paclobutrazol on the flavor and quality of wine during fermentation. Sensor evaluation of wine treated with triazole pesticides was conducted by electronic nose and tongue analyses. Additionally, GC-IMS was used to analyze the differences in the volatile organic compounds (VOCs) of the triazole-pesticide-treated wine samples. SPME-GC-MS was further used to identify and quantify the flavor components present in the tested wine samples. The results from this study provide more accurate information on the wine flavor and quality changes induced by triazole pesticides.

## 2. Results and Discussion

### 2.1. Electronic Sensory Evaluation

As new bionic sensing technologies for smell and taste, electronic nose and tongue sensory systems are simple, fast, nondestructive, and repeatable. Thus, they can provide an alternative to smell and taste evaluation to effectively distinguish the differences between the test samples.

#### 2.1.1. Electronic Nose Analysis

The taste quality of the blank and triazole-pesticide-treated samples was analyzed by electronic nose technology. Principal component analysis (PCA) was applied to the data analysis as a multivariate method of generating principal component (PC) variables by investigating the correlation among several variables [23], which was used to eliminate the correlation among original characteristic variables. According to the PCA analysis (Figure 1a), the first component (PC1) explained 99.76% of the total system variance. This indicated that the first principal components represent most of the valid information of the original data, which can be used to reflect the changes in the wine odor. The flavor difference between any two groups was >0.5 (Appendix A), indicating that there was a significant difference in flavor between the four groups. This may be due to the presence of pesticide residue during fermentation causing significant changes in the flavor profile of all the wine samples; similar results have been reported in the literature [24,25]. In addition to the influence of pesticide residues on yeast fermentation, wild yeast on grape surface has relatively high invertase activity, which may also affect the volatile composition and taste of grape. *Saccharomyces cerevisiae* × *S. kudriavzevii* hybrids are prized for their unique flavor profiles in beer and wine, because these hybrids have good enological properties, such as high glycerol content, decreased ethanol, improved taste, and a lower production of undesirable acetic acid [26]. On the contrary, because of the complexity of the yeast strain, hybrids and introgressed strains from *S. eubayanus* and *S. uvarum* could create an odor, which is considered a brewery contaminant [27]. Linear discriminant analysis (LDA) was used as a dimensionality approach that retains most of the information in the original data and finds the best linear fit that separates two or more groups of samples. From Figure 1b, we can see that the data collection points of the same group of samples were gathered in the same area, while the data collection points of the different groups of samples (three triazole-pesticide-treated experimental groups and control group) were scattered in different areas. Figure 1b also reveals that the volatile odors of the wine samples treated with different triazole pesticides were significantly different in discriminant function 1 (DF1) and discriminant function 2 (DF2), and the four wine samples could therefore be effectively discriminated.

#### 2.1.2. Electronic Tongue Analysis

The electronic tongue analysis system consists of seven flavor sensors, namely AHS (sour), CTS (salty), NMS (umami), and SCS, ANS, CPS, and PKS (general purpose) sensors. In this study, the seven flavor sensors responded to the taste of the four wine samples with different sensitivities. Thus, the taste and quality of the control and triazole-treated wine samples were compared and analyzed by electronic tongue technology. Discriminant factor analysis (DFA) is a useful pattern recognition technique for multivariate data analysis [28], which was used as a supervised linear pattern recognition algorithm for data classification. Electronic tongue DFA (Figure 2a) revealed that the four wine groups were significantly different, indicating marked differences in the tastes of the four groups (Appendix A). The data revealed significant differences between any two groups of each sample (*p* < 0.01), while the flavor of the DXZ wine was different from those of the other three groups. Figure 2b is the radar image of the taste characteristics of the wine treated with triazole fungicides and illustrates the similarity of the response values of each sensor. The data indicate that the most affected flavor was that of the DXZ wine, while the flavors of the SZT and WZC wines were less affected. Notably, paclobutrazol is a plant growth regulator that can regulate the growth and development of crops and induce stress resistance in plants [29]. Studies have shown that this triazole can regulate secondary metabolite contents such as Vitamin C (acerbity) and soluble sugars in fruits, thereby affecting their nutrition and quality [30].

### 2.2. GC-IMS Analysis

#### 2.2.1. Effects of the Different Triazole Pesticide Treatments on the VOCs in Wine

The VOCs in the wine samples comprising the differently treated grapes are illustrated in Appendix A. The color represents the concentration of the substance, whereby white indicates a low concentration and red indicates a high concentration. Additionally, a darker color implies a higher concentration. The contrast model was adopted to select the control (CK) spectra as reference, and those of the other samples were deducted from the reference. Thus, for two identical VOCs, the background after deduction was white, while red and blue backgrounds indicated that the substance concentrations were respectively higher and lower than that of the reference. The results reveal that the volatile component content of the SZT wine was only slightly different from that of the control group, while those of the WZC and DXZ wine samples were significantly different. Appendix A is a PCA analysis chart of all the samples, wherein the peak intensities of all the characteristic peaks are selected as characteristic variables for the PCA. This graph visually indicates the differences between the different samples. Thus, a short distance between the samples represents a small difference, while a long distance represents a significant difference. Appendix A therefore indicate that the VOCs in the WZC and DXZ wines were remarkably similar, while the CK wine was relatively more similar to the SZT wine.

The NIST and IMS databases built in the application software identified 40 signal peaks according to the retention time index of the standard substances and the standard drift time; 23 known components and 17 unknown components were determined by comparison with existing databases. Thus, to further investigate the changes in the main volatile substances, the signal peaks of all the VOCs were selected to form a fingerprint for comparison.

#### 2.2.2. Comparison of the VOC Fingerprints in the Triazole-Pesticide-Treated and Control Wine Samples

Figure 3 presents the gallery plot (fingerprint) of the VOCs in the four groups of wines. Each row in the figure represents all the signal peaks selected in a wine sample, while each column represents the signal peaks of the same VOC in the different wine samples. The plot therefore provides the complete VOC information of each sample as well as the differences in the VOCs of the different samples. In Figure 3a, the component contents in region A were higher in the CK and SZT wines, and mainly comprised propionic aldehyde, isoamyl acetate, ethyl propionate, ethyl isobutyrate, and acetone. On the other hand, the component contents in region B were higher in the WZC and DXZ wines and included ethyl octanate, ethyl caproate, 1-butanol, and ethyl butyrate. In region C, the component contents of the control group were significantly different from those of the treatment groups, in which the contents of components no. 36 and no. 40 decreased after treatment, while the contents of components such as isobutyraldehyde increased after treatment.

Samples containing similar VOCs were also compared. Thus, the components in region D (Figure 3b) comprised a higher content in the control and included acetone, ethyl hexanoate, ethyl octoate, ethyl butyrate, 1-butanol, and isobutyl acetate. Higher levels were found in the SZT wine (region E), which included isobutyraldehyde, propionaldehyde, ethyl isobutyrate, isoamyl acetate, and ethyl propionate. Figure 3c reveals that there is little difference between the VOCs of the WZC and DXZ wines. Only the component contents in region F are slightly higher in the DXZ wines, which include isobutyl acetate, isopropyl ethyl butyrate, propyl acetate, isoamyl acetate, ethyl butyrate, and ethyl propionate.

These results indicated that the main volatile substances in the treated wine include ethyl hexanoate, isobutyl acetate, ethyl isobutyrate, propyl acetate, isoamyl acetate, ethyl propionate, ethyl butyrate, 1-esters (e.g., butanol and ethyl lactate), acetone, propionaldehyde, and isobutyraldehyde. The residues of pesticides may affect the uptake of microorganisms and delay the alcohol fermentation, but esters and aldehydes are still the main volatile components [24]. Similar results have also been reported by other research groups [31,32], which suggested that the ethyl esters produced during alcohol fermentation contribute to the typical fruit aroma of wine. On the other hand, alcohols do not affect the wine flavor quality due to their higher sensory thresholds [33]. Studies have also shown that changes in the esters’ concentrations effect the quality of red wine by altering the fruity aroma [34]. In addition to these esters, other compounds that do not necessarily exhibit fruity aromas may have an important effect on the overall fruity aroma of the wine [35].

### 2.3. Head-Space SPME-GC-MS Analysis

The volatile components of the wine samples were further qualitatively and quantitatively analyzed by GC-MS. The results were compared with the spectra of unknown volatile compounds using the NIST11 database and semi-quantified by the internal standard method. A 20 μL/L cyclohexanone content in the wine samples was used as an internal standard to semi-quantify the content of the volatile substances in the different triazole-pesticide-treated wine samples.

#### 2.3.1. GC-MS Qualitative Analysis

The total ion chromatogram of the volatile compounds of the four wine samples are displayed in Figure 4, whereby a total of 58 volatile substances were detected in the four differently treated wines (Table 1). These comprised 14 alcohols, accounting for 24.1% of the total volatile components; 33 esters (56.9%); six acids (10.3%); a ketone (1.7%); an aldehyde (1.7%); and other components [pentane, 2,4-di-tert-butylphenol, and 1,3-di-tert-butylbenzene; 5.3%]. Thus, the results revealed that alcohols, esters, acids, and to a lesser extent aldehydes and ketones were the main volatile components of the wine samples.

In this study, 47, 45, 46, and 54 volatile compounds were detected in the CK, SZT, WZC, and DXZ wine samples, respectively. Among them, 36 volatile substances were common to the four wine samples, including 22 esters, three acids, nine alcohols, one phenol, and one alkane (Appendix A). Of these, some were alcohols (isoamyl alcohol, 1-octanol, citronellol, and phenylethyl alcohol), esters (isoamyl acetate; ethyl caproate; methyl octanoate; ethyl caprylate; ethyl nonanoate; ethyl caprate; methyl salicylate; ethyl salicylate; ethyl laurate; ethyl myristate; and 3-methylbutyl octanoate), and fatty acids (n-decanoic acid and octanoic acid). Esters have been reported to be the main volatile substance contributors in wine [16]. This is consistent with the GC-IMS results attained in this study (Section 2.2.2).

#### 2.3.2. GC-MS Quantitative Analysis

##### Alcohols

Figure 5 displays the alcohols present in the differently treated wines, with total alcohol contents in the CK, SZT, WZC, DXZ wine samples of 136.6, 135.5, 121.46, and 118.23 mg/L, respectively. Rapp and Mandery [40] proposed that a small amount of the major alcohols has a positive effect on the wine quality, while a total concentration of >300 mg/L endows the wine with an unpleasant taste. For the WZC and DXZ wines, the alcohol contents were significantly reduced, indicating that tebuconazole and paclobutrazol affect the flavor of wine by affecting the fermentation of *S. cerevisiae* [41]. It may be that the level of expression of genes involved in alcohol synthesis is affected, for example, phenylalanine metabolism, lysine degradation and biosynthesis in *S. cerevisiae* are inhibited. Linalool and myristyl alcohol, which have lily and citrus aromas, respectively, were also detected in the WZC wine; nevertheless, since their concentrations were below the odor thresholds, it was assumed that these two compounds did not affect the flavor of the wine. Isobutanol was not detected because the biosynthesis of valine, the precursor amino acid of alcohols, may be affected by the pesticide residues [42]. Additionally, pesticide treatment reduced the content of n-hexanol (C6 alcohol) in the WZC wines, while the higher alcohol and geraniol (terpene) contents were not affected. These results are consistent with those reported by Noguerol-Pato et al. [43]. The concentration of isoamyl alcohol (fusel oil, floral descriptor), 1-octanol (orange fragrance), citronellol (rose fragrance), and phenylethyl alcohol (rose fragrance) were higher than their respective odor thresholds, indicating a significant contribution to the wine flavor. Notably, in this study, the concentration of citronellol decreased under paclobutrazol treatment. This was not consistent with the citronellol concentration changes reported by Oliva et al. [44], which did not change following treatment with six fungicides. This difference was ascribed to the disparate mechanisms of paclobutrazol as a plant growth regulator.

##### Esters

The total ester contents in the CK, SZT, WZC, and DXZ wine samples were 232.84, 239.17, 233.55, and 210.69 mg/L, respectively. The contents in the SZT wine were slightly higher than those of the CK and WZC wine samples. On the other hand, after paclobutrazol treatment, the ester content was significantly reduced. The formation of acetate esters is highly dependent on enzyme activity. These enzymes are responsible for both the synthesis and the hydrolysis of medium-chain fatty acid ethyl esters [45]. The levels of ester were significantly reduced in paclobutrazol-treated wine, which may be related to reduced enzyme activity. The most abundant compound was ethyl caproate (78.35–85.87 mg/L), followed by ethyl caprylate (39.4–55.15 mg/L) and ethyl caprate (20.24–26.95 mg/L); the contents of the other esters were <20 mg/L. The concentrations of isoamyl acetate (banana fragrance); decanoic acid, ethylene ester (coconut fragrance); acetic acid, 2-phenylethal ester (sweet fragrance); and dodecanoic acid, ethylene ester (apricot fragrance) in the pesticide-treated groups were lower than those in the CK wines. On the other hand, the concentrations of isopentyl hexanoate, methyl salicylate, ethyl palmitate, ethyl phenylacetate, ethyl heptanoate, and ethyl lactate were higher or appeared in the treatment group, possibly due to the type of nitrogen composition that the pesticides may confer on the must [46]. Notably, although these esters had a fruity aroma, the threshold was not high or <1. In all the treated wine groups, the ethyl caproate, methyl octanoate, and ethyl caprylate concentrations were significantly higher than their odor thresholds (0.08, 0.2, and 0.51 mg/L, respectively) and significantly differed from the concentrations of the control group. However, a high ester concentration, with a strong fruit flavor, in the treated wine has a negative effect on the aromatic quality of the wine [42].

##### Diverse Volatile Compounds

The total acid substance contents in the CK, SZT, WZC, and DXZ wine samples were 16.97, 16.44, 18.23, and 18.35 mg/L, respectively. Triadimefon slightly influenced the concentration of n-decanoic acid, octanoic acid, and acetic acid. On the other hand, treatment with tebuconazole increased the n-decanoic (unpleasant) and octanoic (smell of rancid butter) acid concentrations, while treatment with paclobutrazol increased the octanoic and hexanoic acid concentrations, which impart the wine with an unpleasant flavor. Only aldehyde, 2-undecenal was detected in the CK wines, giving the wine its fresh aldehyde flavor because of its polar threshold. Thus, treatment with the tested pesticides affected aldehyde synthesis, possibly because they promoted the synthesis of the corresponding alcohols during fermentation [47].

### 2.4. Combined Sensory, GC-IMS, and SPME-GC-MS Analysis

In this study, electronic nose and tongue technologies were used to investigate the flavor of the wine samples treated with different triazole pesticides. The results revealed significant changes in the overall flavor of the wines treated with triazole pesticides, which could be effectively discriminated by electronic nose and tongue technologies.

GC-IMS and GC-MS methods were further used to identify and quantify the volatile components of the wine samples after different treatments. Comparison of the GC-IMS fingerprints indicated that esters are important factors in determining the wine quality. Moreover, because their concentrations are usually higher than their threshold levels, they endow the wine with a fruity aroma [46]. Differences in the types and relative contents of the volatile substances were observed in the different samples. In all, 40 typical compounds were determined by GC-IMS, but there were still 17 compounds with no qualitative results due to the limited data of the library database. Based on the identified compounds, the volatile compounds in samples were mainly esters, alcohols, and aldehydes, which was consistent with the results of the SPME-GC–MS analysis. Of the 58 compounds identified by SPME-GC-MS technology, the main volatile components were esters, alcohols, acids, and some aldehydes, ketones, and alkanes. Paclobutrazol was the most influential pesticide on the volatile components of wine, significantly changing the concentrations of citronellol; isoamyl acetate, hexanoic acid; ethyl ester, octanoic acid; metallic ester, nonanoic acid; and ethical ester hexanoic acid. Tebuconazole and paclobutrazol reduced the ester and alcohol contents in the wine samples, while conversely, the triadimefon-treated samples retained most of their original wine flavor quality. This result was similar to that of the GC–IMS analysis. Oliva et al. [41] reported that fungicides of the triazole family can affect the amount of ethyl esters, acetates, acids, and ethyl acetate in wine. Moreover, they inhibit the synthesis of major sterols on fungal cell membranes, reducing the fermentation activity of *S. cerevisiae* and, consequently, affecting the synthesis of volatile compounds by other metabolic pathways [48,49].

## 3. Material and Methods

### 3.1. Materials and Reagents

Cyclohexanone analytical standards (purity ≥99.9%) were purchased from Dr. Ehrenstorfer (LGC Standards, Augsburg, Germany). Commercial triadimefon 20% emulsifiable concentrate was sourced from Jiangsu Sword Agrochemicals Co., Ltd. (Yancheng, China); tebuconazole 43% suspension was acquired from Qingdao Haina Biotechnology Co., Ltd. (Qingdao, China); and paclobutrazol 15% wet-table powder was obtained from Jiangsu Kesheng Group Co., Ltd. (Yancheng, China). Sodium chloride (NaCl) was provided by Sinopharm Chemical Reagent Co., Ltd. (Shanghai, China). *Saccharomyces cerevisiae* powder was purchased from Yantai Di Boshi brewing machine Co., Ltd. (Yantai, China). The fermentation tanks were bought from Hebei Chaoya glass products Co., Ltd. (Hebei, China). Ultra-pure water was produced using a Millipore purification system (Millipore, Bedford, MA, USA). The grapes were obtained from Changxinghongyuan grape professional cooperative of Liaoning province and did not contain the target pesticides.

### 3.2. Red Wine Processing

A total of 60 kg fresh and mature kyoho grapes was used, the brix was 16° and total acidity was 5 mg/g of berry, with the rotten grape particles and stems removed, and divided into 4 groups, and every 5 kg of grapes was selected and crushed into fermentation vessels; the pulp dregs, juice, and skin were all included in the fermentation tanks. The spiked levels of triadimefon, tebuconazole, and paclobutrazol were 0.3, 2.0, and 0.5 mg/kg, respectively, according to the corresponding maximum residue limits established by GB 2763-2019, and a different pesticide was added to each fermentation tank. Next, 30 mg/kg SO_2_ and 1 g/kg yeast powder activated in 100 mL of 30 °C pure water were added into each fermentation tank, and the mixtures were stirred three times daily at the initial stage of the fermentation process. After 240 h at room temperature (25 °C), the skin residues were filtered out and secondary fermentation was performed during 192 h. Finally, the fermented grape musts were clarified and filtered by sieve to attain four groups of wine samples, bottled at 4 °C for further study; three repetitions were performed for each group. The four groups were labeled CK, representing the control group, and SZT, WZC, and DXZ representing the wines treated with triadimefon, tebuconazole, and paclobutrazol, respectively. The detailed processing procedure is illustrated in Figure 6.

### 3.3. Electronic Nose Detection

A PEN3.5 electronic nose (Airsense Analytics, GmBH, Schwerin, Germany) was employed for the volatile analysis. Thus, 1.0 mL each of the control and three triazole-pesticide-treated wine samples were separately placed into four 10 mL headspace sampling vials. Each sample was injected manually with a PEN3.5 electronic nose and allowed to stand for 10 min, under laboratory conditions of 25 ± 1 °C, until the volatile gases in the sample filled the headspace space of the vials. Electronic nose detection comprised respective sampling and cleaning times of 60 and 180 s and three parallel samples were set for each group.

### 3.4. Electronic Tongue Detection

An Alpha Astree II electronic tongue system connected with an LS16 auto sampler unit (Alpha MOS, Toulouse, France) was used for the interacting chemical substances in solution. Thus, 10 mL aliquots of the control and triazole-pesticide-treated wines were collected, filtered with 0.45 μm aqueous phase filter membranes, and placed into separate 100 mL beaker with a constant volume of 120 mL. Each beaker was then placed in the spot tongue detector sample position so that it was sitting just over the electrode surface. Respective collection and cleaning times of 120 and 10 s were employed. To eliminate the interference of the unstable factors of the initial detection response signal and obtain a stable detection signal, seven parallel measurements were conducted in the study. After obtaining the analysis data, the first three circles of data were discarded, and the last four circles of stable electronic tongue response data were selected as the analysis data.

### 3.5. GC-IMS Analysis of the Volatile Compounds

A commercial GC-IMS (FlavourSpec^®^; GAS, Dortmund, Germany) was used for flavor analysis. Thus, a 1 mL wine sample was first transferred into a 20 mL headspace bottle. Then, 100 μL aliquots were automatically injected under splitless mode at 85 °C, after incubating at 60 °C and 500 rpm incubation speed for 10 min. The separation was carried out on an MXT-WAX column (dimensions, 30 m × 0.53 mm × 1 μm) at 60 °C. Nitrogen (purity, >99.99%) was used as the carrier gas, and the linear pressure program was as follows: 2 mL/min for 2 min, ramped up to 100 mL/min over 18 min, and finally maintained at this rate for 10 min. The drift gas flow rate was 150 mL/min (nitrogen), and a 5 cm drift tube was operated at a constant voltage of 400 V/cm at 45 °C.

### 3.6. Head-Space SPME-GC-MS Analysis of the Volatile Compounds

A gas chromatography-mass spectrometer (Shimadzu GC-MSQP 2010plus) was employed for compound separation and analysis. Thus, 5 mL wine samples were accurately measured into 15 mL headspace bottles, and 1 g NaCl was added to each bottle to promote volatilization of the volatile components. Subsequently, 50 μL cyclohexanone internal standard solution (2 μL/mL) was added and the bottles were immediately sealed with a polypropylene cap comprising a PTFE-silicone rubber septum. The headspace volatile components of the wine samples were then extracted by 65 μm divinylbenzene/polydimethylsiloxane (DVB/PDMS). After equilibration of each sample at 50 °C for 20 min, the extraction head was inserted into a headspace bottle for 40 min. Next, the DVB/PDMS fiber was placed in a 250 °C gas chromatography-mass spectrometer inlet for 2 min to allow desorption. The instrument was equipped with a DB-WAX column (dimensions, 30 m × 0.25 mm × 0.25 μm). Analytical conditions were set as follows: an initial column chamber temperature of 40 °C, injection port temperature of 250 °C, spitless injection mode, and a carrier gas flow rate of 1.0 mL/min. The column chamber temperature was initially set at 40 °C for 3 min, then increased to 120 °C at 5 °C /min, and finally to 230 °C at 10 °C /min where it was held for 5 min. The ion source and transfer-line temperatures were 200 and 250 °C, respectively, and the signal was collected under scan mode in the scanning range 35–500 *m*/*z*. Peak identification was primarily achieved using Lab Solution software in retention index calibration mode with retention index calibrated in-house MS libraries, and compared to the NIST mass spectral search program (NIST 11).

### 3.7. Data Statistics

The wine fermentation experiments were carried out in triplicate, and PCA was used for sample discrimination and classification. LDA was based on the determination of the linear discriminant functions (DFs) to extract features by maintaining class separability. DFA was used to further expand the differences between the different groups and narrow the differences within each group based on principal component analysis. The stable intervals selected for the PCA and LDA analyses were in the range 48–52 s. These analyses were used to distinguish the smell characteristics of the four groups of wine samples. Discriminant factor analysis (DFA) was conducted to compare the differences in the taste characteristics of the four wine sample groups. The instrumental analysis software includes LAV (Laboratory Analytical Viewer, version 2.2.1—G.A.S. Dortmund, Germany) and three plugins (Reporter, Gallery Plot, and Dynamic PCA) as well as GC × IMS Library Search, which can be used for sample analysis from different perspectives.

## 4. Conclusions

The main objective of this study was to explore the influence of three different triazole pesticides on wine flavor. Analysis at the molecular sensing level was based on multidimensional analysis technology. The pesticide with the greatest influence on wine flavor composition was paclobutrazol, as the wines treated with this pesticide exhibited the most significant changes in the concentrations (relative to threshold) of the flavor components (citronellol, isoamyl acetate, ethyl caproate, methyl octanoate, ethyl caprylate, ethyl caprate, methyl cis-4-decenoate, ethyl laurate, ethyl myristate, and hexanoic acid). Treatment with the three pesticides mainly changed the concentrations of the esters, the main contributors to the significant differences in the wine flavors. The study demonstrated that pesticide residues on the grapes are transferred from the grape berries to the grape juice, thereby changing the flavor components and thus, the flavor quality of the final wine. Electronic nose and tongue analyses were also employed to assess the potential flavor differences caused by the triazole pesticides. Moreover, the combination of dynamic headspace sampling with GC-MS and GC-IMS allowed us to determine the compounds present in the volatile components of the wine. The results of the three tests were in agreement, which was conducive to the comprehensive analysis of the effects of the three triazole pesticides on wine flavor. Further, the application of these techniques can effectively avoid the risk of exposing traditional human sensory evaluators to chemical hazards present in target samples. Meanwhile, it is important to monitor the presence of pesticides in grapes and wines in order to improve wine flavor, and biopesticides, which are safe, less toxic, least persistent, and environmentally friendly to humans and non-target organisms, can be used as alternatives to chemical pesticides in the vineyard.

## Figures and Tables

**Figure 1 molecules-25-05596-f001:**
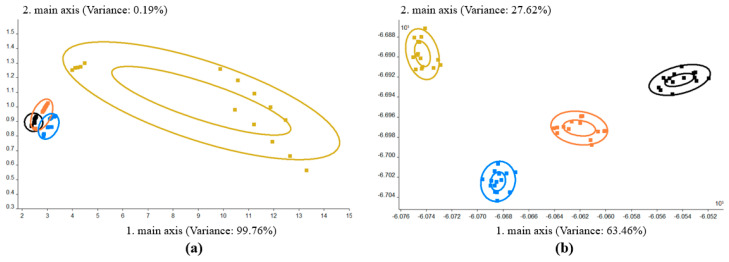
Results obtained by using the electronic nose. (**a**) Principal component analysis (PCA) and (**b**) linear discriminant analysis (LDA) of the electronic nose data for the control (CK; black), triadimefon-treated (SZT; orange), tebuconazole-treated (WZC; blue), and paclobutrazol-treated (DXZ; yellow) wine groups.

**Figure 2 molecules-25-05596-f002:**
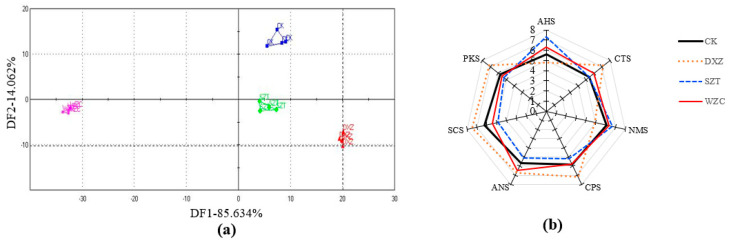
Results obtained by using the electronic tongue. (**a**) Discriminant factor analysis (DFA) diagram of the electronic tongue data for the control (CK; blue), triadimefon-treated (SZT; green), tebuconazole-treated (WZC; pink), and paclobutrazol-treated (DXZ; red) wine groups. (**b**) Radar image of the taste characteristics of the triazole-pesticide-treated wine samples.

**Figure 3 molecules-25-05596-f003:**
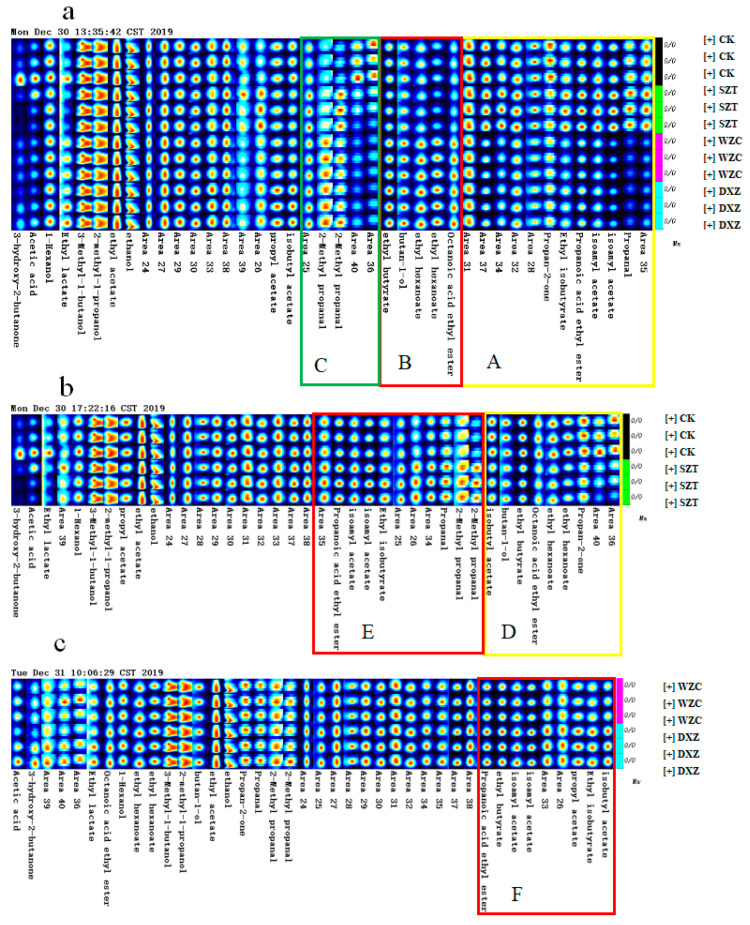
Analyzed by GC-IMS. (**a**) Comparison of gallery plot in all wine samples (CK, SZT, WZC, and DXZ wine samples). (**b**) Comparison of gallery plot between the control (CK) and triadimefon-treated (SZT) wine groups. (**c**) Comparison of gallery plot between the tebuconazole-treated (WZC) and paclobutrazol-treated (DXZ) wine groups. A, B, C, D, E, F represent regions where volatile compounds differ significantly.

**Figure 4 molecules-25-05596-f004:**
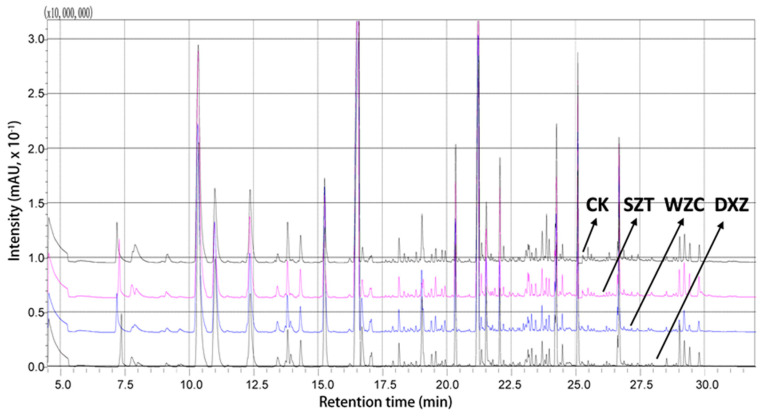
SPME-GC-MS total ion chromatogram of the volatile compounds for, from top to bottom, the blank (CK), triadimefon-treated (SZT), tebuconazole-treated (WZC), and paclobutrazol-treated (DXZ) wine samples.

**Figure 5 molecules-25-05596-f005:**
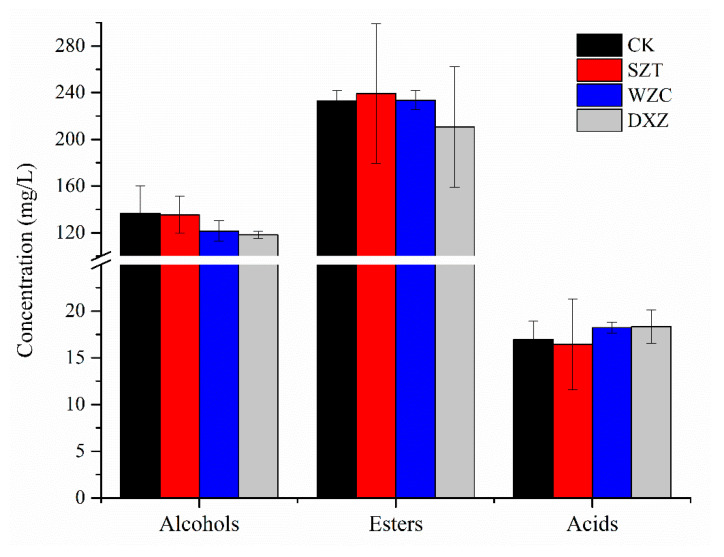
Comparison of relative volatile components of the four wine samples. Blank (CK), triadimefon-treated (SZT), tebuconazole-treated (WZC), and paclobutrazol-treated (DXZ) wine samples.

**Figure 6 molecules-25-05596-f006:**
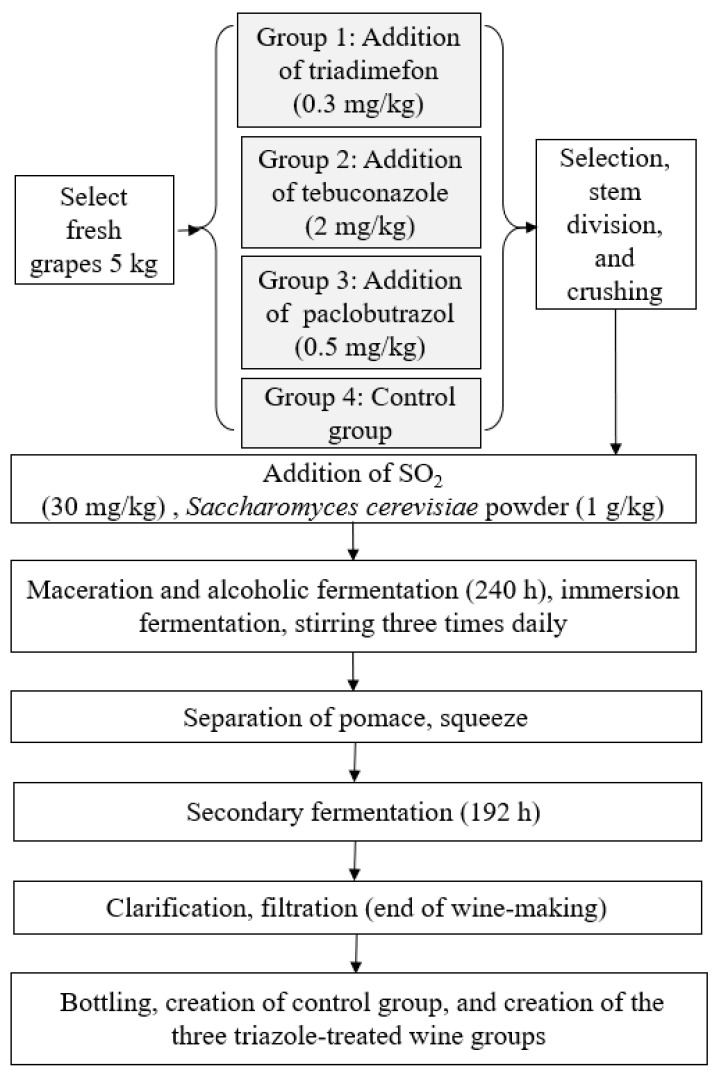
Wine making process adopted in the experiment.

**Table 1 molecules-25-05596-t001:** Composition of the wine volatile substances determined by SPME-GC-MS analysis.

Elution Order	Compound	Retention Index	Descriptor	Concentration (mg/L)				Odor Threshold (mg/L)
				CK	SZT	WZC	DXZ	
1	Ethanol	463	Pungent, wine ^a^	33.13	27.66	22.2	27.59	100 ^d^
2	Isobutanol	597	Disagreeable, wine ^a^	5.27	2.08	--	4.17	40 ^d^
3	Pentane	518	Alkane ^b^	2.48	2.41	7.09	2.33	4.1 ^d^
4	Isoamyl acetate	820	Banana ^a^	6.05	3.39	2.39	1.22	0.2 ^d^
5	Isoamyl alcohol	697	Fusel oil, pungent ^a^	72.82	80.02	73.73	60.65	30 ^d^
6	Ethyl caproate	984	Fruit ^a^	20.46	20.24	20.55	26.95	0.08 ^d^
7	Cyclohexanone *	891	Peppermint ^a^	20.0	20.0	20.0	20.0	20
8	Ethyl heptanoate	1083	Pineapple ^b^	--	--	0.27	0.39	0.3 ^d^
9	Ethyl lactate	848	Wine, cream ^a^	--	--	0.11	1.92	8 ^d^
10	Hexyl formate	981	Apple, unripe plum ^b^	1.28	1.22	2.39	1.1	6.4 ^d^
11	1-Hexanol	860	Sweet, green fruity ^b^	2.77	2.89	1.23	2.34	8 ^d^
12	Methyl octanoate	1083	Oranges, grapes ^a^	13.33	14.8	16.33	18.78	0.2 ^d^
13	1,3-Di-tert-butylbenzene	1334	Unknown	0.33	0.27	--	--	Unknown ^e^
14	Ethyl caprylate	1183	Orange, oily ^a^	85.87	92.28	88.98	78.35	0.51 ^c^
15	Acetic acid	576	Pungent, vinegar ^a^	2.7	1.99	2.53	0.56	22 ^d^
16	Isopentyl hexanoate	1218	Banana, pineapple ^b^	0.82	0.94	1.43	0.96	0.9 ^d^
17	Methyl nonanoate	1183	Wine, coconut ^a^	0.24	0.37	--	0.18	0.04 ^d^
18	Propyl caprylate	1282	Spice ^a^	0.47	--	0.85	0.35	Unknown ^e^
19	Ethyl nonanoate	1282	Grape, rose, wine ^b^	5.24	8.29	2.31	2.84	0.377 ^d^
20	(*R*,*R*)-2,3-Butanediol	743	Unknown	1.3	1.36	--	1.25	Unknown ^e^
21	Linalool	1082	Lily, citrus ^a^	--	--	0.46	--	5 ^d^
22	Isobutyl caprylate	1317	Unknown	0.78	0.76	0.86	0.78	Unknown ^e^
23	1-Octanol	1059	Orange, rose ^a^	1.36	1.91	1.8	1.33	0.13 ^d^
24	2,3-Butanediol	743	Spices ^b^	1.97	1.52	1.6	1.29	20 ^d^
25	Methyl n-caprate	1282	Unknown	10.83	9.7	11.13	8.44	Unknown ^e^
26	Ethyl caprate	1381	Coconut ^a^	55.15	53.01	49.81	39.4	2.4 ^c^
27	Methyl cis-4-decenoate	1290	Unknown	0.57	--	--	0.4	Unknown ^e^
28	Isoamyl caprylate	1417	Fruity ^b^	2.11	--	--	4.77	0.125 ^d^
29	3-Methylbutyl octanoate	1417	Fruity, brandy ^a^	6.73	7.6	7.7	6.3	0.125 ^d^
30	2-Methyl butyric acid	811	Cheese, fruit ^a^	--	--	--	0.31	5.9 ^d^
31	Diethyl succinate	1151	Faint, pleasant ^a^	0.37	0.28	3.09	0.44	200 ^d^
32	3-Methylthiopropanol	912	Onion, meat ^a^	0.33	0.28	0.37	0.26	4 ^d^
33	Ethyl undecanoate	1481	Coconut ^a^	0.41	0.27	--	0.08	Unknown ^e^
34	2-Undecenal	1311	Fresh aldehyde ^a^	0.15	--	--	--	0.001 ^d^
35	1-Decanol	1258	Flower, fatty ^a^	1.69	2	2.1	1.3	2.8 ^d^
36	Citronellol	1179	Rose ^b^	1.26	1.8	1.43	0.7	0.1 ^d^
37	Methyl salicylate	1281	Ilex leaf ^a^	0.96	1.59	1.12	1.45	0.071 ^d^
38	Ethyl phenylacetate	1259	Rose ^a^	--	1.03	1.16	0.96	0.65 ^c^
39	Methyl laurate	1481	Fatty, floral ^a^	2.93	2.36	2.73	2.11	Unknown ^e^
40	Ethyl salicylate	1380	Wintergreen ^b^	0.54	0.6	0.52	0.47	0.115 ^d^
41	Phenethyl acetate	1259	Sweet ^a^	4.17	3.25	1.41	0.87	1.8 ^d^
42	Hexanoic acid	974	Stink, unpleasant ^a^	--	--	--	2.28	0.42 ^d^
43	Ethyl laurate	1580	Apricot ^a^	14.43	12.47	12.93	9.13	0.5 ^d^
44	Isoamyl decanoate	1615	Unknown	1.26	1.47	1.68	0.98	5 ^d^
45	Phenylethyl alcohol	1136	Rose ^b^	14.7	13.98	16.36	17.31	7.5 ^c^
46	1-Dodecanol	1457	Fatty, waxy ^a^	--	--	--	0.04	0.073 ^d^
47	Methyl tetradecanoate	1680	Onion, honey, orris ^a^	0.16	0.12	0.26	0.3	0.5 ^d^
48	Strawberry furanone	1022	Fruit, caramel ^a^	0.12	0.12	0.08	--	Unknown ^e^
49	Ethyl myristate	1779	Essence ^b^	1.27	1.22	1.29	1.86	0.5 ^d^
50	Octanoic acid	1173	Mildly unpleasant ^a^	11.39	11.82	12.25	12.62	10 ^d^
51	Ethyl pentadecanoate	1878	Unknown	--	--	--	0.07	Unknown ^e^
52	Nonanoic acid	1272	Light fat, coconut ^a^	--	--	--	0.08	3.0 ^d^
53	Myristyl alcohol	1656	Essence ^a^	--	--	0.18	--	5 ^d^
54	Methyl hexadecanoate	1878	Unknown	0.3	0.29	0.27	0.46	2 ^d^
55	Methyl palmitoleate	1886	Unknown	0.11	--	0.19	0.06	Unknown ^e^
56	Ethyl palmitate	1978	Mild, sweet ^a^	1.5	1.94	2.04	2.16	2 ^d^
57	*n*-Decanoic acid	1372	Unpleasant ^a^	2.88	2.63	3.45	2.5	1 ^d^
58	Ethyl 9-hexadecenoate	1986	Unknown	1.23	1.64	1.73	1.25	Unknown ^e^
59	2,4-Di-tert-butylphenol	1555	Burning, sweet ^a^	1.45	2.45	1.82	1.37	Unknown ^e^

The volatile components of the wine samples were semi-quantified by the internal standard method; dashes “--” in the columns denote that the sample was not detected; CK, SZT, WZC, DXZ represent the control and triadimefon-, tebuconazole-, and paclobutrazol-treated wine; ^a^ Reference [36]; ^b^ Reference [37]; ^c^ Reference [38]; ^d^ Reference [39]; ^e^ Unknown odor value; * Internal standard.

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
