# Peer review of "Influence of Triazole Pesticides on Wine Flavor and Quality Based on Multidimensional Analysis Technology"

_molecules, 2020, doi:10.3390/molecules25235596_

Round 1

Reviewer 1 Report

Manuscript titled “Influence of triazole pesticides on wine flavor and quality based on multidimensional analysis technology” studies the effects of triadimefon, tebuconazole, and paclobutrazol on the flavor and quality of wine during fermentation.
Analytical methods as well as all analysis were conducted without doubts. Experimental designed, materials and the realization of the experiment is presented very clearly.
Manuscript is suitable for publication in present form.

Author Response

Reviewers' comments:

Manuscript titled “Influence of triazole pesticides on wine flavor and quality based on multidimensional analysis technology” studies the effects of triadimefon, tebuconazole, and paclobutrazol on the flavor and quality of wine during fermentation. Analytical methods as well as all analysis were conducted without doubts. Experimental designed, materials and the realization of the experiment is presented very clearly. Manuscript is suitable for publication in present form.

Response:

We are very grateful to the editor and reviewers for the effort and time spent reviewing our manuscript titled ‘Influence of triazole pesticides on wine flavor and quality based on multidimensional analysis technology’. Thanks for all your feedback and we greatly appreciate the positive comments.

Reviewer 2 Report

Comments for molecules-998676

This is an interesting study, which highlight the negative impact of some pesticide residues (triadimefon, tebuconazole and paclobutrazol) on sensory quality of wines.

Some suggestions were done:

  1. Materials and methods section should be presented at the end, before the Conclusion section.
  2. Figure 2 must be divided in two distinct figures corresponding to the electronic nose analysis (a and b) and electronic tongue analysis (c,d), respectively
  3. A detailed discussion of the statistical approaches must be performed, justifying each time the selection of the statistical approach.
  4. Please justify the presence of section 4. Combined Sensory, GC-IMS, and SPME-GC-MS analysis, in the manuscript

In conclusion, the work could be suitable for publication in Molecules, after some completions and clarifications.

Author Response

Reviewers' comments:

Comments for molecules-998676

This is an interesting study, which highlight the negative impact of some pesticide residues (triadimefon, tebuconazole and paclobutrazol) on sensory quality of wines. Some suggestions were done. In conclusion, the work could be suitable for publication in Molecules, after some completions and clarifications.

Response: Thank you for your comments concerning our manuscript entitled “Influence of triazole pesticides on wine flavor and quality based on multidimensional analysis technology” (ID: molecules-998676). These comments are all valuable and very helpful for revising and improving our paper, as well as the important guiding significance to our research. Based on the comments and suggestions, we have revised the manuscript carefully. Revised portions are marked in red in the manuscript.

Point 1: Materials and methods section should be presented at the end, before the Conclusion section.

Response 1: Thank you for your suggestion. We have now made the corresponding change. Moved the materials and methods section at the end of the article, before the Conclusion section.

Point 2: Figure 2 must be divided in two distinct figures corresponding to the electronic nose analysis (a and b) and electronic tongue analysis (c,d), respectively.

Response 2: Many thanks for your suggestion. We have now divided figure 2 into two distinct figures. The order of the pictures has been adjusted, figure 1 for the electronic nose analysis and figure 2 for the electronic tongue analysis, respectively.

Point 3: A detailed discussion of the statistical approaches must be performed, justifying each time the selection of the statistical approach.

Response 3: Thank you very much for your kind advice to improve our manuscript. Based on your comments and suggestions, we have carefully discussed each statistical method. Revised portions are marked in red in the manuscript.

Point 4: Please justify the presence of section 4. Combined Sensory, GC-IMS, and SPME-GC-MS analysis, in the manuscript

Response 4: Thank you for your constructive suggestion. In section 4, we combined sensory, GC-IMS and SPME-GC-MS analysis. The results of sensory analysis showed that the sensory differences between samples were significant. After that, we analyzed the flavor compounds of the differences by GC-IMS, and finally confirmed and quantified the difference substances by GC-MS. The combined application of these technologies can organically integrate to complement the insufficiency of each other, and also can effectively avoid the risk of chemical hazards existing in the target samples of traditional human sensory evaluators and can accurately and quantitatively analyze the impact of different pesticides on wine flavor.

Reviewer 3 Report

Dear Editor and Authors,

Please, find my comments, recommendations and suggestions on the manuscript below:

Abstract

No comments

Keywords: flavor components; odor activity value – these two phrases contain a similar meaning. Therefore, it is better to remove one of them (maybe the second one) and to include another appropriate keyword describing your study.

Introduction:

Line 50: actinomycetes and basidiomycetes - Could you list some important for the Agriculture microorganisms as examples?

Line 56: cytochrome activity – Which important cytochromes can be affected?

Lines 57058: Thus, it is important to monitor the presence of pesticides and regulate their levels in grapes and wine to limit human health risks. – After this sentence, you can also suggest some alternative biotechnological approaches to use biopesticides (with a reference) as a step to control the problem reported in the study.

Materials and Methods:

No comments

Results and Discussion: Please, try to improve the connection between the chemical composition and sensory characteristic changes suggesting the biochemical changes (on pathways in yeasts) found from the treatment with the respective pesticides.

Lines 197-199: The flavor difference between any two groups was >0.5 (Table S1), indicating that there was a significant difference in flavor between the four groups. This is because the presence of pesticide residue during fermentation caused significant changes in the flavor profile of all the wine samples – What is your comment about that result? Do you expect any effects on the metabolisms of yeasts used or inhibition on wild yeasts found on the grape surface, which also influence the volatile profile and taste? Please, try to explain it.

Lines 278-281: These results indicated that the main volatile substances that significantly affect the flavor of the treated wine include ethyl hexanoate, isobutyl acetate, ethyl isobutyrate, propyl acetate, isoamyl acetate, ethyl propionate, ethyl butyrate, 1-esters (e.g., butanol and ethyl lactate), acetone, propionaldehyde, and isobutyraldehyde. – It is necessary to try to explain the effects on flavor taking into consideration the effects on the metabolism of yeasts (general pathways for synthesis of these compounds?) when pesticides present.

Fig. 5: Could you try to comment the slightly decreased amount of alcohols after treatment with some of the pesticides?  This comment can be valuable for explanation of the results.

Lines 355-356: On the other hand, after paclobutrazol treatment, the ester content was significantly reduced. – Any hypothesis for (a) biochemical pathway(s) affected in yeasts or anything else?

It will be necessary to try to evaluate the negative impact of different pesticides on formation or release from the grapes of valuable biologically active compounds on the basis of the chemical profiles.

Did you find any changes in wine color as a result of the treatment with pesticides? Please, explain that, too (if applicable).  

Conclusion:

Here you need to finish this section with a perspective application of your results or their usefulness for plant protection perspectives for cultures used for wine making.

References:

Please, double check your reference list for any typos. (Line 473: dots for abbreviated words? - Ind J. Plant Physiol.)

Finally, I would like to recommend publication of the manuscript in Molecules after a minor revision as suggested above.

Sincerely yours,

Reviewer

Author Response

Thank you for your comments concerning our manuscript entitled “Influence of triazole pesticides on wine flavor and quality based on multidimensional analysis technology” (ID: molecules-998676). These comments are all valuable and very helpful for revising and improving our paper, as well as the important guiding significance to our research. Based on the comments and suggestions, we have revised the manuscript carefully. Revised portions are marked in red in the manuscript.

Keywords:

Point 1: flavor components; odor activity value – these two phrases contain a similar meaning. Therefore, it is better to remove one of them (maybe the second one) and to include another appropriate keyword describing your study.

Response 1: Thank you for your recommendations. We have now removed the keyword “odor activity value” and changed to “fermentation” in the abstract. And now the revised Keywords are as follows: triazole pesticides; wine; fermentation; sensory analysis; flavor components.

Introduction:

Point 1: Line 50: actinomycetes and basidiomycetes - Could you list some important for the Agriculture microorganisms as examples?

Response 1: Thank you very much for your kind advice. we have listed some important for the agriculture microorganisms as examples in the manuscript: “Like Streptomyces scabies, a genus of actinomycetes which causes scab in tap root crops and potato tubers [6]. And the grape skin extract containing fungicide can obviously inhibit the germination of Penicillium expansum, Penicillium chrysogenum and Aspergillus niger [7].” Line (51-53).

  1. LERAT, S., SIMAO-BEAUNOIR, A.-M., & BEAULIEU, C. Genetic and physiological determinants of Streptomyces scabies pathogenicity. Mol Plant Pathol. 2009, 10, 579–585.
  2. Corrales, M., Fernandez, A., Vizoso Pinto, M. G., Butz, P., Franz, C. M. A. P., Schuele, E., & Tauscher, B. Characterization of phenolic content, in vitro biological activity, and pesticide loads of extracts from white grape skins from organic and conventional cultivars. Food. Chem Toxicol. 2010, 48, 3471–3476.

Point 2: Line 56: cytochrome activity – Which important cytochromes can be affected?

Response 2: Thank you for bringing this to our attention. Triazole pesticides can significantly inhibit the enzymatic activity of cytochrome 3A4 (CYP3A4), which was the dominant form of CYP450 in the liver that mediates the 6b-hydroxylation of testosterone. We have added this information in the revised manuscript (Line 59-61).

Point 3: Lines 57-58: Thus, it is important to monitor the presence of pesticides and regulate their levels in grapes and wine to limit human health risks. – After this sentence, you can also suggest some alternative biotechnological approaches to use biopesticides (with a reference) as a step to control the problem reported in the study.

Response 3: Thank you for your helpful comments. We have now added the following text to discuss this: “And use phytochemistry biopesticides that are less toxic, least persistent, environmentally friendly, and safe to humans and non-target organisms. It was reported that several phytochemical biopesticides like azadirachtin, nicotine, pyrethrins, rotenone, veratrum, annonins, rocaglamides, isobutylamides etc. have been successfully commercilalized in the past [10].” (Lines 62-66).

  1. Walia, S., Saha, S., Tripathi, V., & Sharma, K. K. Phytochemical biopesticides: some recent developments. Phytochem Rev. 2017, 16, 989–1007.

Results and Discussion:

Point 1: Please, try to improve the connection between the chemical composition and sensory characteristic changes suggesting the biochemical changes (on pathways in yeasts) found from the treatment with the respective pesticides.

Response 1: Thank you very much for your kind advice to improve our manuscript. Triazole pesticides such as triadimefon acts mainly by influencing steroid biosynthesis to inhibit cell wall synthesis and steroid biosynthesis, glutathione metabolism, phenylalanine metabolism, and interference sphingolipid metabolism, so as to reduce the fermentation activity of yeast, further affecting other metabolite pathways to impact the synthesis of flavor compounds.

  1. Kong, Z., Li, M., An, J., Chen, J., Bao, Y., Francis, F., & Dai, X. The fungicide triadimefon affects beer flavor and composition by influencing Saccharomyces cerevisiae metabolism. Sci Rep. 2016, 6, 33552.

Point 2: Lines 197-199: The flavor difference between any two groups was >0.5 (Table S1), indicating that there was a significant difference in flavor between the four groups. This is because the presence of pesticide residue during fermentation caused significant changes in the flavor profile of all the wine samples – What is your comment about that result? Do you expect any effects on the metabolisms of yeasts used or inhibition on wild yeasts found on the grape surface, which also influence the volatile profile and taste? Please, try to explain it.

Response 2: Many thanks for your suggestion. We have now modified the text in the manuscript according to your suggestions: “This may be due to the presence of pesticide residue during fermentation caused significant changes in the flavor profile of all the wine samples; similar results have been reported in the literature [23, 24]. In addition to the influence of pesticide residues on yeast fermentation, wild yeast on grape surface has relatively high invertase activity, which may also affect the volatile composition and taste of grape. S. cerevisiae × S. kudriavzevii hybrids are prized for their unique flavour profiles in beer and wine [25]. On the contrary, because of the complexity of the yeast strain, hybrids and introgressed strains from S. eubayanus and S. uvarum could create an odor, which is considered a brewery contaminant [26]” (Lines 122-129).

  1. Peris, D., Pérez-Torrado, R., Hittinger, C. T., Barrio, E. & Querol, A. On the origins and industrial applications of Saccharomyces cerevisiae × Saccharomyces kudriavzevii hybrids. Yeast. 2018, 35, 51–69.
  2. Langdon, Q. K., Peris, D., Baker, E. P., Opulente, D. A., Nguyen, H.-V., Bond, U., et al. Fermentation innovation through complex hybridization of wild and domesticated yeasts. Nat Ecol Evol. 2019, 3, 1576–1586.

Point 3: Lines 278-281: These results indicated that the main volatile substances that significantly affect the flavor of the treated wine include ethyl hexanoate, isobutyl acetate, ethyl isobutyrate, propyl acetate, isoamyl acetate, ethyl propionate, ethyl butyrate, 1-esters (e.g., butanol and ethyl lactate), acetone, propionaldehyde, and isobutyraldehyde. – It is necessary to try to explain the effects on flavor taking into consideration the effects on the metabolism of yeasts (general pathways for synthesis of these compounds?) when pesticides present.

Response 3: Thank you for your suggestion. We have now modified the text to discuss this: The residues of pesticides may affect the uptake of microorganisms and change the fermentation or delay the alcohol fermentation, but esters and aldehydes are still the main volatile components[23]. Similar results have also been reported by other research groups [30, 31], which suggested that the ethyl compounds produced during alcohol fermentation contribute to the typical fruit aroma of wine. Line (216-220)

Point 4: Fig. 5: Could you try to comment the slightly decreased amount of alcohols after treatment with some of the pesticides?  This comment can be valuable for explanation of the results.

Response 4: Many thanks for your helpful comments. We have modified the text in the manuscript according to your suggestions. “For the WZC and DXZ wines, the alcohol contents were significantly reduced, indicating that tebuconazole and paclobutrazol affect the flavor of wine by affecting the fermentation of S. cerevisiae [37]. It may be that the level of expression of genes involved in alcohol synthesis is affected. Such as Phenylalanine metabolism, lysine degradation and biosynthesis in S. cerevisiae are inhibited.” Line (270-272)

Point 5: Lines 355-356: On the other hand, after paclobutrazol treatment, the ester content was significantly reduced. – Any hypothesis for (a) biochemical pathway(s) affected in yeasts or anything else?

Response 5: Thank you for bringing this to our attention. The formation of acetate esters is highly dependent on enzyme activity. These enzymes are responsible for both the synthesis and the hydrolysis of medium-chain fatty acid ethyl esters [41]. And the levels of ester were significantly reduced in paclobutrazol treated wine, which may be related to reduced enzyme activity.

  1. Malcorps, P., Cheval, J. M., Jamil, S., & Dufour, J. P. A new model for the regulation of ester synthesis by alcohol acetyltransferase in Saccharomyces cerevisiae during fermentation. J. Am Soc. Brew Chem. 1991, 49, 47–53.

Point 6: It will be necessary to try to evaluate the negative impact of different pesticides on formation or release from the grapes of valuable biologically active compounds on the basis of the chemical profiles.

Response 6: Thank you for your suggestion. Figure S1 is the structural formula of three pesticides. Pesticides may remain in the wine as parent compounds or as degradation products. The antimicrobial activities of pesticides and their degradation products are different among different pesticides, so they may have different effects on yeasts [7].

  1. Corrales, M., Fernandez, A., Vizoso Pinto, M. G., Butz, P., Franz, C. M. A. P., Schuele, E., & Tauscher, B. Characterization of phenolic content, in vitro biological activity, and pesticide loads of extracts from white grape skins from organic and conventional cultivars. Food. Chem Toxicol. 2010, 48, 3471–3476.

Point 7: Did you find any changes in wine color as a result of the treatment with pesticides? Please, explain that, too (if applicable).

Response 7: Thank you for your comments. We have carefully observed the colour of the wines. The color of wine samples was dark purple and was slightly different with no obvious difference between the treatment and control. We have learned that anthocyanin is the main pigment in red wine. During fermentation, Anthocyanin will change due to the presence of fungicides, which will affect the color of wine [23].

  1. Briz-Cid, N., Castro-Sobrino, L., Rial-Otero, R., Cancho-Grande, B., & Simal-Gándara, J. Fungicide residues affect the sensory properties and flavonoid composition of red wine. J. Food Compos. Anal. 2018, 66, 185–192.

Conclusion:

Point 1: Here you need to finish this section with a perspective application of your results or their usefulness for plant protection perspectives for cultures used for wine making.

Response 1: Many thanks for your suggestion. Based on these comments and suggestions, we have modified the text in the manuscript: “It is important to monitor the presence and management of pesticides in grapes and wines in order to improve wine flavor. And to promote the use of phytochemistry biopesticides, which is safe, less toxic, least persistent and environmentally friendly to humans and non-target organisms, taking into account the impact on human health.” Line (448-451)

References:

Point 1: Please, double check your reference list for any typos. (Line 473: dots for abbreviated words? - Ind J. Plant Physiol.)

Response 1: Thank you for bringing this to our attention. We have revised the relevant information in the revised manuscript (Page 16; Lines 521). We apologize for the oversight. The revised reference list is as follows:

  1. Bhattacherjee, A. K., & Singh, V. K. Uptake of soil applied paclobutrazol in mango cv. Dashehari and its persistence in soil, leaves and fruits. Ind. J. Plant Physiol. 2015, 20, 39–43.